# Haplotype-level metabarcoding of freshwater macroinvertebrate species: A prospective tool for population genetic analysis

**Joeselle M. Serrana**[1,2¤], **Kozo Watanabe**[1]*

**1** Center for Marine Environmental Studies, Ehime University, Matsuyama, Ehime, Japan, **2** Faculty of Engineering, Graduate School of Science and Engineering, Ehime University, Matsuyama, Ehime, Japan

¤ Current address: Ottawa Institute of Systems Biology, and the Department of Biochemistry, Microbiology, and Immunology, Faculty of Medicine, University of Ottawa, Ontario, Canada

* watanabe.kozo.mj@ehime-u.ac.jp

**Data Availability Statement:** The raw sequence data were deposited into the National Center for Biotechnology Information (NCBI) Sequence Read

## Abstract

Metabarcoding is a molecular-based tool capable of large quantity high-throughput species identification from bulk samples that is a faster and more cost-effective alternative to conventional DNA-sequencing approaches. Still, further exploration and assessment of the laboratory and bioinformatics strategies are required to unlock the potential of metabarcoding-based inference of haplotype information. In this study, we assessed the inference of freshwater macroinvertebrate haplotypes from metabarcoding data in a mock sample. We also examined the influence of DNA template concentration and PCR cycle on detecting true and spurious haplotypes. We tested this strategy on a mock sample containing twenty individuals from four species with known haplotypes based on the 658-bp Folmer region of the mitochondrial cytochrome c oxidase gene. We recovered fourteen zero-radius operational taxonomic units (zOTUs) of 421-bp length, with twelve zOTUs having a 100% match with the Sanger haplotype sequences. High-quality reads relatively increased with increasing PCR cycles, and the relative abundance of each zOTU was consistent for each cycle. This suggests that increasing the PCR cycles from 24 to 64 did not affect the relative abundance of each zOTU. As metabarcoding becomes more established and laboratory protocols and bioinformatic pipelines are continuously being developed, our study demonstrated the method's ability to infer intraspecific variability while highlighting the challenges that must be addressed before its eventual application for population genetic studies.

## Introduction

DNA-based techniques for species identification, e.g., Sanger sequencing, have been developed for cases where morphology-based identification proved problematic [1]. DNA barcodes, i.e., selected short fragments of DNA containing genetic information specific to each organism, are used to identify species or higher taxa depending on their level of variability. For example, the typical marker for identifying eukaryotic species is the mitochondrial cytochrome c oxidase I (mtCOI) gene [1]. However, conventional DNA barcoding faces practical limitations

Archive (SRR14631855). The Sanger (GenBank accession numbers also listed in Table 1) and ZOTU sequences (Fasta format), other input data, and the R markdown document implementing the analyses contained in this manuscript are available in the figshare data repository at https://doi.org/10.6084/m9.figshare.15090180.v1.

**Funding:** KW was supported by the Japan Society for the Promotion of Science (JSPS) Grant-in-Aid for Scientific Research (Grant No. 19K21996 and 19H02276). The funders had no role in study design, data collection and analysis, decision to publish, or preparation of the manuscript.

**Competing interests:** The authors have declared that no competing interests exist.

restricting the method to single specimen analysis, making DNA extraction and sequencing of large-scale samples expensive [2]. Hence, conventional approaches do not always fulfill the need of ecologists and are not ideal for large quantity high-throughput species identification [3].

To overcome the limitations of processing large numbers of specimens, and with the recent advancements of molecular technologies, researchers tapped next-generation sequencing technologies to allow DNA barcode-based identification to be conducted in a massively parallel manner [4]. In particular, high throughput sequencing of amplified DNA markers, aka metabarcoding, enables the simultaneous multi-species identification of community samples containing DNA from different origins [3]. The advancements in sequencing platform technologies made it possible to generate millions of sequences from community (i.e., mixed or mass-collected) samples and potentially identify most taxa, including the rare and inconspicuous ones [5–8]. The development and application of metabarcoding for biodiversity surveys have been considered a game-changer for ecological research, specifically for eukaryotic organisms [6]. Other than the taxonomic characterization of the community sample, metabarcoding data can be used to infer biodiversity indices and has been established as a robust method for environmental impact assessments of freshwater ecosystems [e.g., 9–13]. For these applications, biodiversity is generally quantified with taxonomic, functional, or phylogenetic diversity at the community level (i.e., interspecific diversity) [14]. However, intraspecific diversity, the genetic variation within species from community samples, is typically not explored.

Including intraspecific diversity assessment in ecological monitoring and planning management strategies would be beneficial, given that haplotype data are direct proxies for the spatiotemporal dynamics of populations that can be substantially different compared to community-level assessments [3]. In particular, evaluating changes in population size in response to environmental stressors [e.g., 15, 16] are key areas of basic and applied ecological research [17]. Quantifying gene flow between populations can examine the magnitude and mechanisms of population connectivity. Metabarcoding is a faster and more cost-effective alternative to the conventional approach of repeated genetic analysis for multiple individuals from different populations. It may also allow the comparison of population genetic structures of multiple species with different life stages and dispersal modes or abilities [18].

Previous literature has proposed the applicability of metabarcoding for population genetic analysis [e.g., 18–21]. A handful of studies in freshwater environments have also presented the possibility of inferring haplotype information from metabarcoding of individually-tagged specimens [e.g., 22], macroinvertebrate community samples [e.g., 23–26], and environmental DNA collected from water samples [e.g., 27–31]. These studies highlighted the possibility of extracting haplotype or sequence variant information from high-throughput marker-gene sequencing data. However, the high number of reads containing sequencing errors that may occur throughout the metabarcoding procedure, from polymerase chain reaction (PCR) amplification bias and errors to bioinformatics pipelines, may influence the reliability of metabarcoding for population genetic analysis [32, 33]. DNA template concentration and the number of PCR cycles introduce bias and errors, mainly when applied to community samples [34, 35]. This makes it difficult to distinguish true haplotypes from erroneous sequencing noise [33]. Thus, further exploration and assessment of the laboratory and bioinformatics strategies are required to unlock the potential of DNA metabarcoding-based inference of haplotype information.

Using a mock sample with known Sanger-sequenced haplotypes (referred to as true haplotypes here on) of the 658-bp barcode region of the mtCOI gene, we assessed whether these haplotypes would be identified in the metabarcoding data. We also examined the influence of varying DNA template concentration and PCR cycles on detecting true haplotypes and reducing spurious haplotypes obtained from metabarcoding using the unoise3 denoising parameters

[36]. As metabarcoding becomes more established and laboratory protocols and bioinformatics pipelines are continuously being developed, we demonstrated that the method could infer intraspecific genetic variability, showing promising applications for population genetic analysis.

## Materials and methods

### Study species and Sanger-sequence haplotypes

We inferred haplotypes from a mock sample by pooling extracted DNA of 20 individuals from four species with known haplotypes assessed from published population genetics studies (*Amphinemura decemseta*) [37] or our DNA barcoding projects (*Kamimuria tibialis*, *Eucapnopsis bulba*, and *Epeorus latifolium*) (Table 1). Each individual sample was morphologically identified accordingly, and the selected haplotypes for each species had variable nucleotide diversity that would be good for this assessment. Additional information on the Sanger sequence generation of these samples is provided as a supplementary text in S1 File. The haplotypes (*h*), haplotype diversity (*Hd*), and the total number of mutations (*Eta*) were calculated in DnaSP v.6.10.04 [38]. The genealogical relationship of the haplotypes of each species via a median-joining network was determined using a haplotype network tree (Fig 1A) drawn using the PopART software [39]. To examine if DNA template concentration influences the detection of true haplotypes from the mock sample, haplotypes for each species were prepared with

**Table 1. Sample and haplotype information of the individuals used for the mock sample.**

| Order | Family | Species | GenBank Accession | Sample Code | Conc. ng/uL[a] | Haplotype Information | | | | | |
|---|---|---|---|---|---|---|---|---|---|---|---|
| | | | | | | 658-bp[b] | | | 421-bp[c] | | |
| | | | | | | *h* | *Hd* | *Eta* | *h* | *Hd* | *Eta* |
| Plecoptera | Nemouridae | *Amphinemura decemseta* | MK132341 | AD1 | 0.01 | 3 | 1 | 6 | 2 | 0.7 | 5 |
| | | | MK132323 | AD2 | 0.05 | | | | | | |
| | | | MK132340 | AD3 | 0.1 | | | | | | |
| | Perlidae | *Kamimuria tibialis* | MZ543957 | KT1 | 0.01 | 7 | 1 | 245 | 5 | 0.9 | 169 |
| | | | MZ543958 | KT2 | 0.05 | | | | | | |
| | | | MZ543959 | KT3 | 0.1 | | | | | | |
| | | | MZ543960 | KT4 | 0.5 | | | | | | |
| | | | MZ543961 | KT5 | 1 | | | | | | |
| | | | MZ543962 | KT6 | 5 | | | | | | |
| | | | MZ543963 | KT7 | 10 | | | | | | |
| | Capniidae | *Eucapnopsis bulba* | ON678129 | EB1 | 0.01 | 5 | 1 | 104 | 5 | 1 | 71 |
| | | | ON678130 | EB2 | 0.05 | | | | | | |
| | | | ON678131 | EB3 | 0.1 | | | | | | |
| | | | ON678132 | EB4 | 0.5 | | | | | | |
| | | | ON678133 | EB5 | 1 | | | | | | |
| Ephemeroptera | Heptageniidae | *Epeorus latifolium* | MZ543952 | EL1 | 0.01 | 5 | 1 | 19 | 5 | 1 | 14 |
| | | | MZ543953 | EL2 | 0.05 | | | | | | |
| | | | MZ543954 | EL3 | 0.1 | | | | | | |
| | | | MZ543955 | EL4 | 0.5 | | | | | | |
| | | | MZ543956 | EL5 | 1 | | | | | | |

[a]DNA concentration in the mock sample

[b]Haplotypes based on the 658-bp Folmer region of the mtCOI gene

[c]421-bp internal COI fragment of the BF2 + BR2 primers. "*h*" refers to the number of haplotypes; "*Hd*" haplotype diversity; "*Eta*" Total number of mutations.

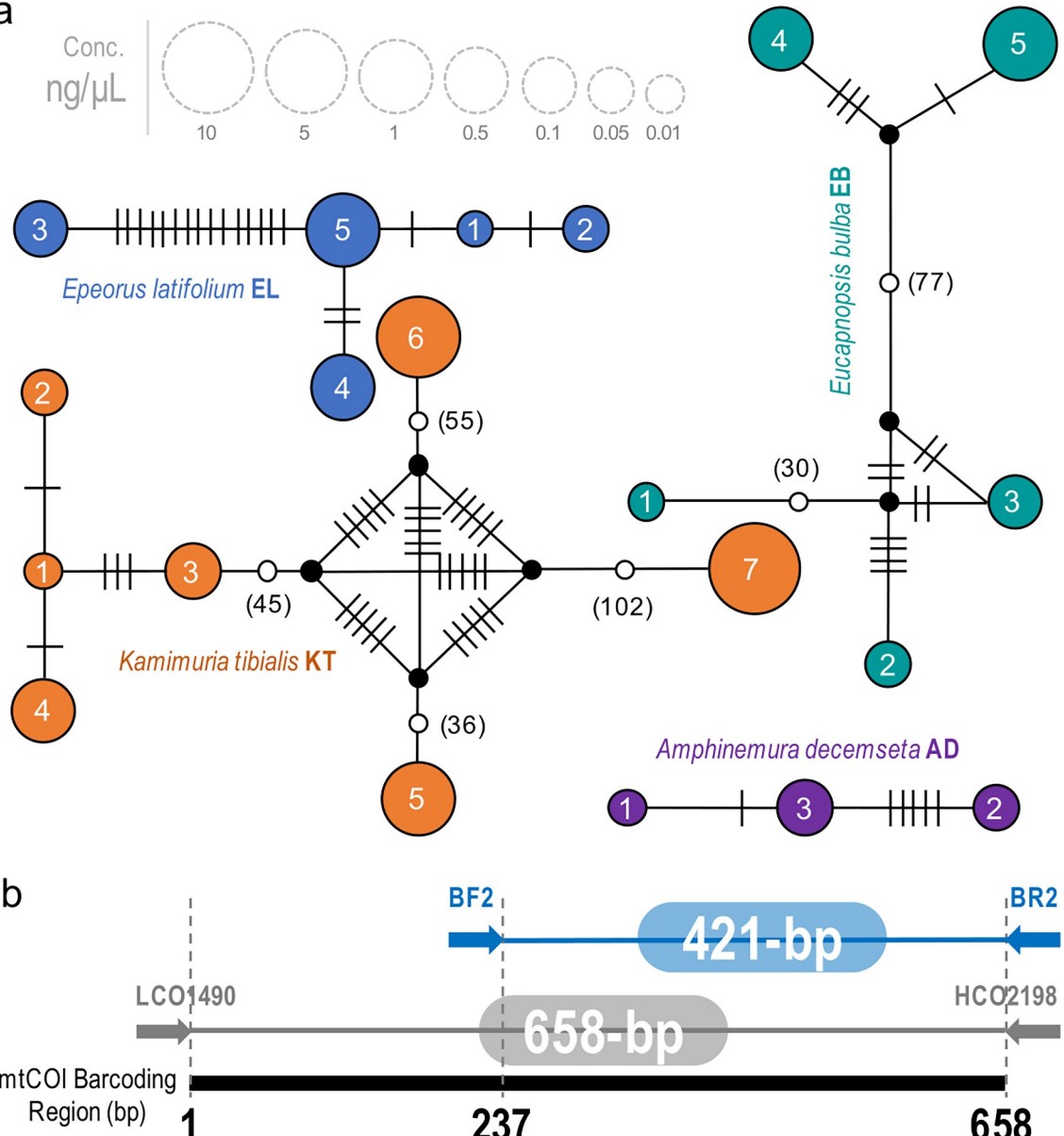

**Fig 1. Haplotype networks and barcode information.** Median-joining haplotype networks showing the level of intraspecific variation within each of the four species based on the 658-bp mtCOI Sanger sequence (a). The circle size represents the concentration of the extracted DNA of each haplotype in ng/μL, i.e., 0.01 (1), 0.05 (2), 0.1 (3), 0.5 (4), 1 (5), 5 (6), 10 (7), and the number of mutations is represented as hatch marks or numbers. Additional two samples with different concentrations of 5.0 and 10.0 ng/μL. Position of the sequenced amplicon (BF2 & BR2) along the mtCOI barcode region (b).

different concentrations before pooling their DNA samples in one tube for amplicon library preparation. The DNA concentration was quantified with the QuantiFluor dsDNA system (Promega, Madison, WI, USA) on the Quantus Fluorometer (Promega, Madison, WI, USA), and sample concentrations from 0.01, 0.05, 0.1, 0.5, and 1.0 ng/μL were prepared (Table 1). The mock sample included two samples with high concentrations of 5.0 and 10.0 ng/μL for *K. tibialis* (Table 1).

## Library preparation and next-generation sequencing

The extracted DNA of the 20 individuals adjusted to their respective concentrations were pooled by adding 15 μl each in one tube. Elbrecht and Leese's [40] fusion primers (BF2 and BR2; Fig 1B) for freshwater arthropods were used to construct amplicon libraries following a one-step PCR protocol. Each primer has unique inline shifts and Illumina adapters attached to the 5'-end for sample tagging and multiplexing. The PCR master mix consists of 0.25 μl Phusion polymerase with five μl HF Buffer (New England Biolabs, NEB), 0.75 μl DMSO, one μl dNTPs, 1.25 μl each of the fusion forward and reverse primers (10 μM), and 15.5 μl of PCR-grade water. PCR cycling conditions were 30 s of initial denaturation at 98˚C, followed by 20 cycles of 10 s denaturation at 98˚C, 30 s annealing at 55˚C, 30 s extension at 72˚C, and a final extension step of 5 minutes at 72˚C. The same PCR protocol was done for eleven other cycles from 24 to 64 (increments of 4) in triplicate PCR amplifications, including no-template (blank) controls for each PCR cycle (also in triplicates).

The 72 PCR samples were pooled according to replicates and size selected via solid-phase reversible immobilization (SPRI) beads. Each replicate was quantified via qPCR with the KAPA Library Quantification Kit (Kapa Biosystems, Wilmington, MA, USA). Before sequencing, quality was assessed via the DNA 1000 assay using the Agilent 2100 BioAnalyzer (Agilent Technologies, Palo Alto, CA, USA). The replicate amplicon libraries were normalized to 2nM before pooling to ensure even read output distribution between replicates. Paired-end sequencing of the pooled library spiked with 20% PhiX was performed on the Illumina MiSeq platform using the MiSeq Reagent v3 600-cycle Kit (2 × 300 cycles).

## Metabarcoding data processing and haplotype inference

The raw Illumina MiSeq paired-end reads were demultiplexed according to sample tags via the R package JAMP v.0.67 [32] and were quality-checked with FastQC [41]. We demultiplexed 14.7 million reads assigned to each of the 72 samples (S1 Table in S1 File). The paired-end reads were merged via the JAMP pipeline, primer stripped, truncated at 421-bp, quality filtered with a maximum expected error filtering value of 2, and dereplicated, excluding singletons using the USEARCH v11.0.667 pipeline [42]. To extract individual haplotypes from the dereplicated sequences, we employed a denoising strategy using unoise3 [36]. The zero-radius operational taxonomic unit (zOTUs) sequences were mapped against the 658-bp mtCOI Sanger sequences of the mock samples using the UPARSE-REF algorithm [42]. The command is designed for validating mock community sequencing experiments where the set of biological sequences in the sample is known. Note that sub-OTUs, OTUs with 100% sequence similarity, and zOTUs are synonymous [36, 43]. Similarly, amplicon sequence variants (ASVs) and exact sequence variants (ESVs) are terms used for outputs of other denoising pipelines (e.g., DADA2 [44]). We used zOTU in this study since we employed unoise3's algorithm for denoising the quality-filtered reads [36].

We performed phylogenetic inference via neighbor-joining analysis of the Sanger and the zOTU sequences employing the Jukes-Cantor substitution model with bootstrapping (1000) via the online MAFFT multiple sequence alignment software version 7 [45]. Before the analysis, the 658-bp Sanger sequences of the mock samples were truncated to the 421-bp length of the BF2 and BR2 barcodes. Data visualization (i.e., boxplots and bubble plots) and statistical analysis were performed in R v.4.2.3 [46]. The default Wilcoxon T-test analysis was performed to compare the read abundances of the raw, merged, and zOTU-assigned sequences between the DNA template and negative controls for each cycle using the function *stat_compare_-means()*. To normalize the read abundance per sample, the zOTU table was log-transformed

using the *phyloseq_standardize_otu_abundance()* function in the metagMisc v0.5.0 package [47] for phyloseq objects generated using the phyloseq v.1.42.0 package [48].

# Results

## Sanger sequence haplotypes

The mock sample consisted of four different species, i.e., *Amphinemura decemseta* (Nemouridae), *Kamimuria tibialis* (Perlidae), *Eucapnopsis bulba* (Capniidae), and *Epeorus latifolium* (Heptageniidae) from two freshwater insect orders (Table 1). After trimming the Sanger sequences into the 421-bp length of the BF2 and BR2 barcode region to complement the amplified region in the DNA metabarcoding data, three of the *K. tibialis* Sanger haplotypes (i.e., KT1, KT3, and KT4) were grouped into one haplotype, similar to the two haplotypes of *A. decemseta* (i.e., AD1 and AD3). Hence, the 20 haplotypes from the 658-bp long fragment of the mtCOI barcoding region [49] were reduced to 17 haplotypes after trimming to 421-bp length.

## Read abundance, reference sequence match, and phylogenetic inference

The read abundances significantly differed between the samples and blank controls, except for the raw and merged read counts of samples at 20 cycles (Fig 2A). From the demultiplexed reads, 90% were merged, but only 3% were retained after denoising (S1 Table in S1 File). Most of the reads removed were filtered out being erroneous sequences after the denoising step. From the dereplicated reads, 462,665 were assigned to 14 zOTUs (also referred to as "DNA metabarcoding haplotypes" in this study) (S2 Table in S1 File) of 421-bp length. For assessing the different PCR cycle numbers, all of the zOTUs were detected from cycles 24 to 64. For 20-cycle, only eight of the 14 zOTUs were represented, all of which match haplotypes with high concentrations (i.e., KT7, KT6, KT5, EL5, EL4, EB5, EB4, and EB3) (Fig 2B). Quality passing reads increased with increasing cycles, and the relative abundance of each zOTU was relatively consistent across the cycle numbers (S2 Table in S1 File). Notably, four zOTUs were detected in the negative samples from different cycles. zOTU04 was represented on nine negative samples from cycles 28 to 64, zOTU09 and zOTU11 on two negative samples, and zOTU08 and zOTU10 on one, i.e., cycles 60 and 64, respectively. However, it should be noted that most of these occurrences in the negative samples were singleton among samples or doubleton among samples and reads, with one sample (i.e., 40BR1) having the highest detection of only 22 reads.

After mapping the zOTU sequences against the 20 Sanger sequence-haplotypes with 658-bp, 12 zOTUs, comprised of 450,793 (97%) matched-to-zOTU reads, had 100% sequence match against 12 of the Sanger haplotypes of the mock reference sequences (S3 Table in S1 File). The remaining eight Sanger haplotypes that were not detected from the metabarcoding dataset were the *A. decemseta* samples (AB1, AB2, and AB3), two *E. bulba* (EB1 and EB2), *E. latifolium* (EL1), and two *K. tibialis* (KT1 and KT3). Based on the neighbor-joining tree (Fig 3), the remaining zOTUs (i.e., zOTU10 and zOTU14) without a 100% taxonomic match against the Sanger sequences clustered with the *A. decemseta* sequences.

# Discussion

Using a mock sample with known haplotypes based on Sanger sequencing of the highly variable mtCOI gene region, we presented the feasibility and limitations of using metabarcoding data to extract intraspecific genetic diversity information by denoising the sequences into zOTUs. Most recent studies that inferred intraspecific diversity from macroinvertebrate

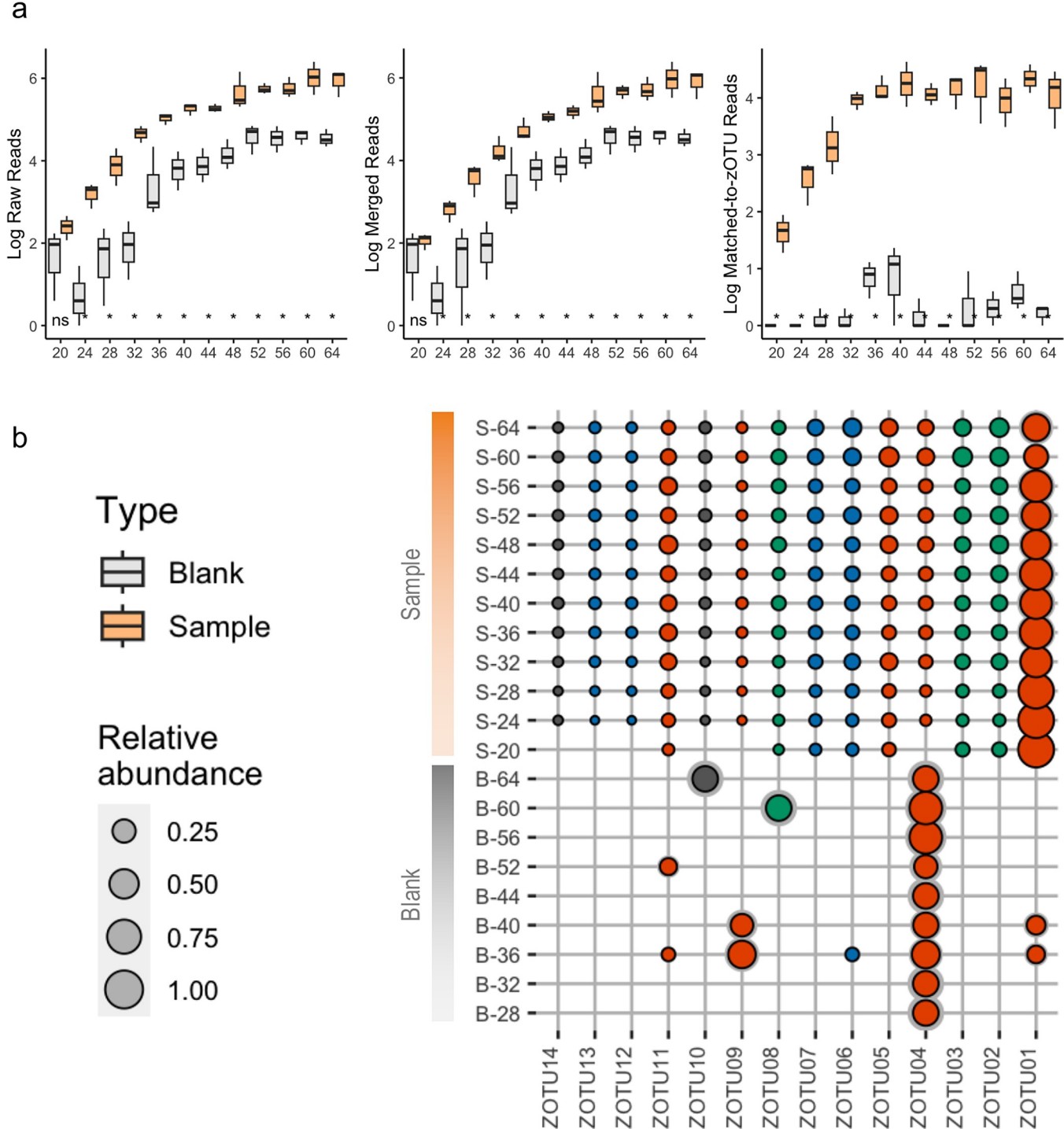

**Fig 2. Read processing and zero-radius operational taxonomic units (zOTU) abundance.** Comparison of read abundance (log-transformed) at different read processing steps per cycle and between the DNA template samples and the negative controls (a). Significance code: '**' associated with a variable at $p < 0.01$ and '*' at $p < 0.05$. Bubble plot showing the relative abundance of zOTUs per PCR cycle (b). Circle size represents the Mean values of three replicates, and the gray shadow is the standard deviation (SD).

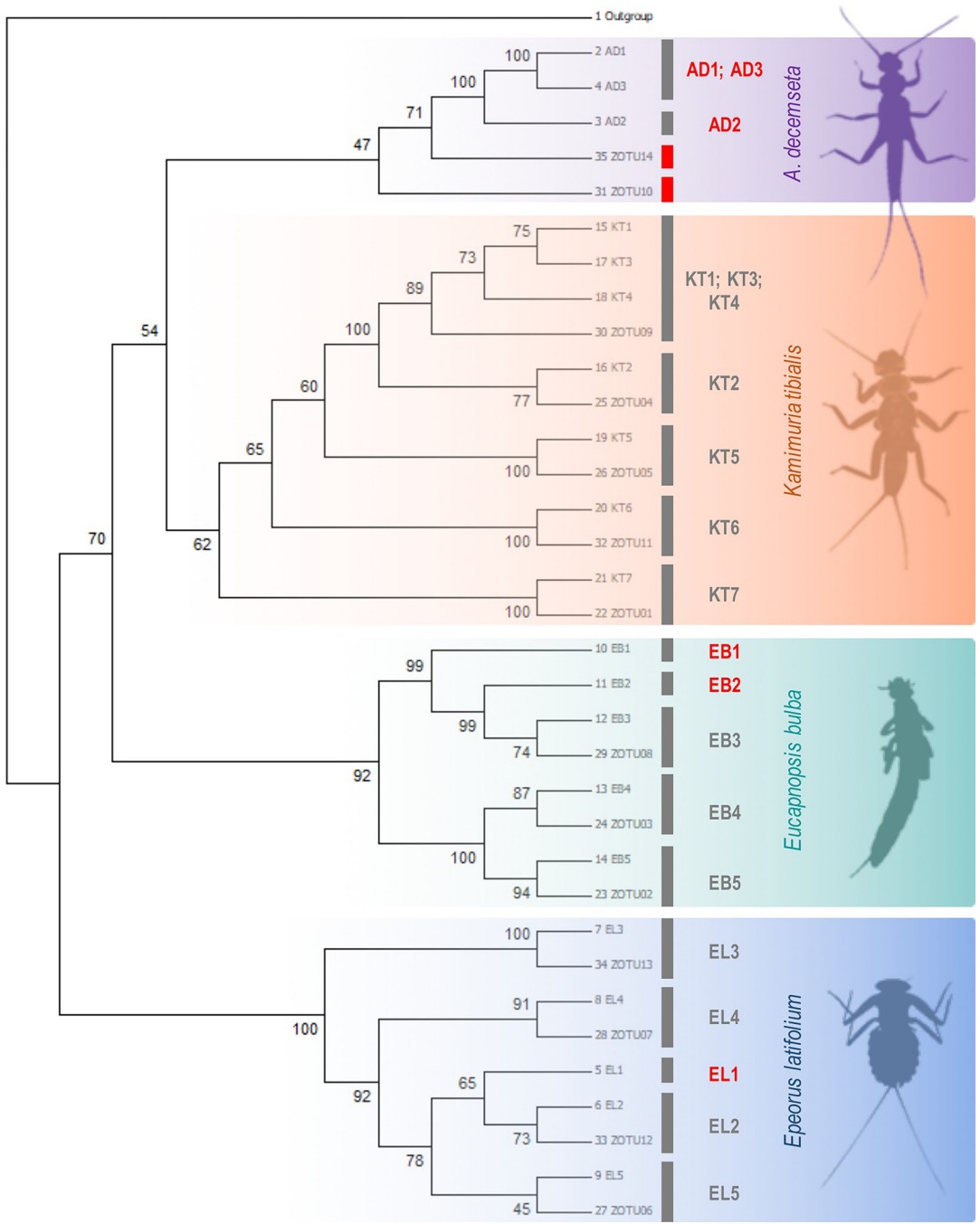

**Fig 3. Neighbor-joining tree of the mock and the zero-radius operational taxonomic units (zOTU) sequences.** Sequences are highlighted based on species. The red bar represents a zOTU without a taxonomic match, and the text in red represents a Sanger haplotype without a zOTU match.

metabarcoding data were also based on denoising algorithms [e.g., 32, 33, 50, 51]. Before denoising, our read processing steps, i.e., paired-end merging, stripping adapter and primer sequences, barcode length truncation, quality filtering, and dereplication, were performed mainly following default and stringent settings. It is also important to note that we truncated and only retained merged reads with a 421-bp (i.e., the entire length of the BF2 and BR2

barcode region) to prevent the generation of shorter-than-barcode zOTUs for a full-length match against the trimmed Sanger haplotype sequences.

## DNA metabarcoding for haplotype-level inference

Some species lost nucleotide information, i.e., polymorphic sites, to distinguish the haplotypes due to shortening the barcode region (658-bp to 421-bp). For example, the *K. tibialis* haplotype with a relatively high input DNA template concentration (i.e., 0.10 ng/μL) was likely not a false negative detection. Its reads were present in the DNA metabarcoding data but merged with the other *K. tibialis* haplotype. This highlights the inability of shorter marker regions to differentiate haplotypes of certain species, which requires further development and evaluation. However, many high-throughput sequencing platforms used for metabarcoding still have strict limitations in sequence length, not utilizing the entire length of the mtCOI barcode [4]. Opportunely, recent advancements in sequencing technologies to generate longer sequences could allow the full-length generation of longer DNA barcodes from community samples [52], which may resolve the limitations of short amplicon sequencing in metabarcoding. Longer reads would allow the generation of high-resolution mitochondrial haplotype data, with potential applications for demographic history and selection analyses [53]. Still, given the raw read error rate from these long-read sequencing platforms, i.e., around 6% for quality scores at least equal to 10 for the Oxford Nanopore MinION[TM] [54] (Delahaye & Nicolas, 2021), further assessment and exploration of library preparation and error-correction methods are recommended [53] for its utilization for metabarcoding studies, more so for haplotype-level inference.

The two metabarcoding haplotypes that failed to have a perfect sequence match against the Sanger sequences clustered with the *A. decemseta* samples. Although we could identify them as *A. decemseta* sequences based on the phylogenetic approach, metabarcoding failed to detect a 100% match of the Sanger haplotypes of this species from the mock sample. We could not rule out PCR amplification, primer bias, or sequencing errors as the reason for these false positive detections or spurious haplotypes obtained from the metabarcoding data, even if we performed relatively strict read quality filtering parameters. Identifying and eliminating artificial or false haplotypes has been a major challenge for population-level inference from metabarcoding data. Elbrecht et al. [32] (2018) reported that artificial or false haplotypes could not be entirely excluded even with stringent filtering settings due to undetected chimeric sequences or systematic sequencing errors that might persist across replicates. Macé et al. [55] (2022) suggested that a denoising method with an additional bimeric sequence removal step combined with a specific polymorphic mitochondrial barcode might resolve the issue of false haplotype detection. In addition, read filtering by relative abundance could help remove false positives and chimeras, given that these sequences are usually low in abundance [55]. Some studies have performed additional read filtering steps based on haplotype presence rate for different PCR replicates [28, 29]. However, these previous resolutions might lead to removing true haplotypes in the community sample with lower abundance. Hence, additional assessment and development of read processing are warranted moving forward.

## Effects of PCR condition on the inference of haplotypes in a mock sample

Initial DNA template concentration significantly influenced the detection of individuals from a mock sample. This observation is in accordance with previous studies, which reported that samples or taxa with low DNA template concentrations had lower detection probability [56]. Accordingly, abundant taxa or samples with high biomass tend to have higher detection probabilities than the rare, smaller, or low-biomass individuals from mixed-community samples

[57–59]. The difference in biomass affects haplotype detection since most of the large specimens would be retained after read processing. These factors must be addressed when metabarcoding-based haplotyping is used to infer abundance-based analysis for population genetics applications.

Additionally, we used a PCR annealing temperature of 55˚C chosen after a temperature gradient PCR of each individual, which was also the optimal condition used from a previous bulk metabarcoding study on freshwater macroinvertebrate communities [13]. This temperature is relatively higher than the typical conditions, i.e., 50˚C, previously used for the BF2 and BR2 fusion primers [e.g., 60, 61]. This could have also led to the non-detection of the low-concentration haplotypes (e.g., AD1, EB1, EL1 with 0.01 ng/uL) in the mock sample. Although previous studies have reported that annealing temperatures from 40–56˚C did not universally affect taxonomic recovery [62], lowering the annealing temperature could have recovered the haplotypes with low DNA template concentration [9]. Hence, we recommend further assessment of the PCR conditions expected to influence species recovery from metabarcoding samples, e.g., GC content [63], annealing temperature [64], polymerase [65, 66], and other primer sets that match the region of interest for freshwater macroinvertebrate species.

The quality-passed reads relatively increased with increasing cycles, and the relative abundance of each zOTU was consistent for each cycle number. This suggests that increasing the PCR cycle from 24 to 64 did not affect the relative abundance of quality-passed reads of each zOTU. Our findings align with previous studies that reported minor or no effect of PCR cycle number on amplification bias [63, 67]. This contrasts with other reports that increasing PCR cycles reduces the proportion of sequences with low starting DNA or less well-amplified species in the mock sample [68]. Moreover, higher PCR cycles have been reported to increase the formation of chimeric sequences and amplification bias [9]; that is why some metabarcoding protocols discourage increasing the PCR cycles above 30. However, a literature survey on bulk sample metabarcoding studies showed that 73% of reports used more than 30 PCR cycles to circumvent primer annealing issues or amplify samples with low amounts of DNA [69]. Our findings proved otherwise, wherein low template samples were undetected or not amplified even after increasing the cycle number to 64 cycles. The relative abundances of the samples detected were relatively consistent for each cycle. Nonetheless, we note that the sequence diversity in our mock sample was relatively low (i.e., four species with 20 haplotypes). A more diverse community might present a different pattern in PCR cycle effects; hence, it warrants further evaluation. Also, primer design is a major issue for metabarcoding studies, and our report was limited to using one fusion primer.

Properly selecting the DNA marker in a metabarcoding assay is crucial because all downstream analyses, e.g., species detection and identification, rely on the marker's ability to amplify and discriminate the representative taxa of the target organisms [4]. The mtCOI gene has a relatively high mutation rate and can detect intraspecific variation. Thus, its widespread use for population genetics studies [e.g., 70, 71], and has been widely used for DNA metabarcoding macroinvertebrates to date. With this, intraspecific variation can be extracted from community samples using various algorithms for sequence clustering and phylogenetic rates [62, 72, 73]. Here, we showed the possibility of generating haplotype information from metabarcoding the mtCOI gene in a mock sample based only on a single gene marker, which could be enough to test or derive population-level hypotheses, e.g., taxa dispersal and distribution at unprecedented scales [32]. Still, given the relatively low cost of metabarcoding, multi-marker assessments can be employed for a more comprehensive population genetic assessment of mixed community samples.

Moreover, nine blank samples had sequences for some of the zOTUs. The presence of these reads in the blank samples might be due to tag jumps or the amplification of sequences

carrying false combinations of used nucleotide tags that is common for dual-indexed libraries [60, 74–76]. Here, we generated amplicon libraries using a one-step PCR strategy with fusion primers developed for freshwater arthropods [40]. This strategy would produce lesser false combinations of tags on the samples compared to a two-step PCR approach. Although we observed tag-jumping for some of the zOTUs, most of these occurrences were singleton or doubleton reads, which could be removed from the downstream analysis. Still, the challenges with tag jumping and contamination between libraries require attention to alleviate false read-to-sample assignments, which would be problematic once the method is employed with environmental samples.

## Conclusion

We demonstrate that metabarcoding can infer intraspecific variability and confirm its ability to detect true haplotypes with the classical Sanger method as a basis, showing promise for possible applications in population genetic studies. In particular, we showed that haplotype information could be extracted from mixed community samples of freshwater macroinvertebrates. Quality-passed reads relatively increased with increasing PCR cycle numbers. However, the relative abundance of each zOTU was consistent across the cycle numbers suggesting that increasing the cycles did not affect the relative abundance of quality-passed reads in this low-diversity mock sample. Although conventional population genetics tools, e.g., Sanger sequencing, are used for targeted sequencing of specific genes or regions, metabarcoding is advantageous for studying complex mixtures since it enables high-throughput sequencing allowing massively parallel analysis of many samples. Hence, metabarcoding has a lower per-sample cost than Sanger sequencing [77, 78]. Still, the overall cost would depend on the number of samples and the sequencing depth required. As metabarcoding becomes more established and laboratory protocols and bioinformatics pipelines are continuously being developed, our study demonstrated that the method could be used to infer intraspecific variability, showing promise for possible applications and highlighting the challenges that need to be addressed before haplotype-level metabarcoding can entirely be used for population genetic applications. This includes further assessment of the laboratory, e.g., amplicon library constriction, PCR conditions, and sequence read processing approaches, e.g., denoising and read filtering steps, needed to confidently recover intraspecific information from metabarcoding data.

## Supporting information

**S1 File.**
(DOCX)

## Acknowledgments

We thank Dr. Dávid Murányi, Dr. Maribet Gamboa, and Dr. Sakiko Yaegashi for the DNA samples and their molecular data. We also thank Dr. Naohito Tokunaga of the Division of Analytical Bio-Medicine for his assistance in performing high-throughput sequencing on the Illumina MiSeq platform of the Advanced Research Support Center (ADRES) at Ehime University, Japan.

## Author Contributions

**Conceptualization:** Joeselle M. Serrana, Kozo Watanabe.

**Data curation:** Joeselle M. Serrana.

**Formal analysis:** Joeselle M. Serrana.

**Funding acquisition:** Kozo Watanabe.

**Methodology:** Joeselle M. Serrana, Kozo Watanabe.

**Project administration:** Kozo Watanabe.

**Supervision:** Kozo Watanabe.

**Validation:** Joeselle M. Serrana, Kozo Watanabe.

**Visualization:** Joeselle M. Serrana.

**Writing – original draft:** Joeselle M. Serrana.

**Writing – review & editing:** Kozo Watanabe.

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
