## [Decision Letter · Decision Letter 0]

16 May 2023

PONE-D-23-02290Haplotype-level metabarcoding of freshwater macroinvertebrate species: a prospective tool for population genetic analysisPLOS ONE

Dear Dr. Watanabe,

Thank you for submitting your manuscript to PLOS ONE. After careful consideration, we feel that it has merit but does not fully meet PLOS ONE’s publication criteria as it currently stands. Therefore, we invite you to submit a revised version of the manuscript that addresses the points raised during the review process.

First of all, I would like to apologize for the considerable delay with this submission. I was invited to handle the manuscript at the end of March, and it has been difficult to find reviewers. Fortunately, three experts in the field agreed to evaluate your work. Their assessments are overall positive and very constructive, but also highlight a number of points that deserve your attention. 

Overall, I would like to encourage the authors to pay attention to comments related to technical aspects, which, if addressed, will help improving the manuscript. This includes clarification and/or rephrasing of statements regarding preparation of mock communities, high ‘loss’ of sequences in the pipeline, justification for thermal conditions and for very small mock community with low taxonomic diversity, as well as the discussion of some of the results (further discussion why some of the haplotypes in the mock community were not detected, and how to overcome this issue; future directions). As such, it would be helpful to include a closing paragraph summarizing the perspectives and recommendations to further develop the approach.

Finally, I recommend the authors pay special attention to concerns raised by reviewer 2 regarding the motivation of this study. While novelty and/or perceived impact of a study is not a criterium for publication in PlosONE, haplotype-level metabarcoding approaches have recently been explored in the referenced study by Elbrecht et al. (2018). In this light, I agree with the reviewer and recommend a slight reframing of the overall study focus (e.g., development of a more time and cost-effective method using NGS to barcode large numbers of bulk samples). Framed in this context, the study would benefit by reaching a broader audience and increasing its visibility.

Please submit your revised manuscript and a detailed point-to-point response by Jun 30 2023 11:59PM. If you will need more time than this to complete your revisions, please reply to this message or contact the journal office at plosone@plos.org. Please include the following items when submitting your revised manuscript:A rebuttal letter that responds to each point raised by the academic editor and reviewer(s). You should upload this letter as a separate file labeled 'Response to Reviewers'.A marked-up copy of your manuscript that highlights changes made to the original version. You should upload this as a separate file labeled 'Revised Manuscript with Track Changes'.An unmarked version of your revised paper without tracked changes. You should upload this as a separate file labeled 'Manuscript'.

We look forward to receiving your revised manuscript.

Kind regards,

Claudia Isabella Pogoreutz

Academic Editor

PLOS ONE

“KW was supported by the Japan Society for the Promotion of Science (JSPS) Grant-in-Aid for Scientific Research (Grant No. 19K21996 and 19H02276).”

3. Thank you for stating the following in the Acknowledgments/ Funding Section of your manuscript:

“This work was supported by the Japan Society for the Promotion of Science (JSPS) Grant-in-Aid for Scientific Research (Grant No. 19K21996 and 19H02276).”

“KW was supported by the Japan Society for the Promotion of Science (JSPS) Grant-in-Aid for Scientific Research (Grant No. 19K21996 and 19H02276).”

Reviewers' comments:

Reviewer's Responses to Questions

**Comments to the Author**

1. Is the manuscript technically sound, and do the data support the conclusions?

Reviewer #1: Yes

Reviewer #2: Yes

Reviewer #3: Yes

2. Has the statistical analysis been performed appropriately and rigorously? 

Reviewer #1: Yes

Reviewer #2: Yes

Reviewer #3: Yes

3. Have the authors made all data underlying the findings in their manuscript fully available?

Reviewer #1: Yes

Reviewer #2: Yes

Reviewer #3: Yes

4. Is the manuscript presented in an intelligible fashion and written in standard English?

Reviewer #1: Yes

Reviewer #2: Yes

Reviewer #3: Yes

5. Review Comments to the Author

Reviewer #1: The presented paper aims to evaluate whether haplotype information can be obtained from DNA metabarcoding studies for population genetic analyses. For that, the authors used a mock community of known haplotypes and tested the influence of input DNA concentration, and the number PCR cycles on the detection ability of the known haplotypes. Interestingly, they found that the relative abundance OTUs were stable with increasing number of PCR cycles. The detection of a few haplotypes failed, which is probably due to their low input DNA concentration. The manuscript is well written, and the data and analyses results support the drawn conclusions. All data is available, and links with analysis scripts are functional. Some points need to be addressed and clarified before the paper can be accepted, but in general the study is scientifically sound and therefore, I recommend it for publication after minor revision.

Introduction:

Line 50: Please correct it to ‚is typically not explored’.

Methods:

Line 104: Please rephrase this sentence. You did not prepare different concentrations of each haplotype; each haplotype was present exactly one time in a specific concentration, and the 20 haplotypes were present in varying concentrations. Reading this sentence, I had the expectation that several mock communities were prepared, which is not the case.

Line 112-114: These fusion primers have an inline shift, which allow sample tagging. I assume these were used and samples were tagged at this step, otherwise you could not demultiplex 72 after pooling and sequencing. Please add this information for readers convenience.

Line 117: These are not the PCR cycling conditions, which are typically used for BF2/BR2. Why did the authors choose this temperature regime? The higher annealing temperature makes the reaction more specific i.e., mismatches between primer and DNA template are less tolerated. This step of the PCR is a competition of molecules. Can the increased annealing temperature affect the detection rate of low concentration haplotypes?

Line 133: Please add the citation of FastQC: “Andrews, S. (2010). FastQC: A Quality Control Tool for High Throughput Sequence Data”

Line 142-145: I would slightly disagree with the authors here. Although conceptually very similar, ASVs and OTUS clustered at 100% sequence similarity are not necessarily the same. ASVs/ESVs are the output of a pipeline which includes denoising. ZOTUs could be also just sequences clustered at 100% similarity and this is much less strict than denoising. Since the authors applied a denoising step, I would suggest using the term ASV instead of ZOTUs, since this is the more appropriate term for many people. However, I understand why the term ZOTU was selected and will leave it to the authors to decide that.

Line 155: What exactly is a metabarcoding frequency? I think this should be rather the frequencies of the ZOTUs.

Results:

Line 172: ~500.000 reads assigned to the target ZOTUs over all samples is a relatively low read number, considering the ~15 million reads in total. What happened to the remaining reads? Are these filtered out during bioinformatic processing, in particular denoising? Or are these reads related to contamination? With increasing cycle number, the number of reads in the blanks can be at the order of 10.000. Although this is a considerable amount of contamination, I understand that it is of less concern since the target ZOTU read abundances are very low (if present at all). Please add a statement which explains why you had only ~500.000 target reads in the end.

Discussion:

Line 202-204: I don’t understand this sentence because the ZOTUs are shorter than the barcode region.

Line 233-236: I agree to this but want to ask the authors whether the lower detection probability of the ‘rare’ haplotypes could be further reduced by the higher annealing temperature (see comment above) used in this experiment?

Line 267-273: Although this is true, there are many studies out there showing that COI over or underestimates 'true' species diversity (in freshwater macroinvertebrates) when compared to nuclear loci (mito-nuclear discordance). I know that this is a methodological study, but I think it would be beneficial to add a short statement about this issue. Do the authors think that true population genetic analyses can be performed based on a single marker system? For example, K. tibialis showed very high intraspecific diversity based on COI, with a genetic distance of more than 15% for individual haplotypes. Is anything known whether this a cryptic species complex? Or is this rather an artefact of historical isolation, which remains in the mitochondrial genome despite recent gene flow?

Figures:

Fig. 1: I don’t see the benefit of adding the concentration information in the scale of the circle size AND the number again. This information is much easier to understand from Table 1. I suggest adding the haplotype ID instead of the ‘concentration number’ to the network.

Fig. 2b: This figure has a very low resolution, and the sample abbreviations are hard to read. Why are not all blanks included? Further I would propose to plot first all ‘real samples’ on the left, and then the controls on the right. This would much better show that relative abundances remain stable with increasing cycle number. In the figure description, the word ‘mean’ (line 509) should be lower case.

Supplement:

The header of Table S2 is not the correct header.

Thank you for your paper and all the best!

Reviewer #2: In their study, "Haplotype-level metabarcoding of freshwater macroinvertebrate species: a prospective tool for population genetics analysis," Serrana and Watanabe explore the potential of using DNA metabarcoding to investigate population genetic patterns at the haplotype level. They conducted experiments on a mock community comprising four species and 20 haplotypes, examining the impact of different DNA concentrations and PCR cycles on the recovery of haplotypes and the detection of true and spurious haplotypes. They employed Sanger-sequenced specimens to create Illumina libraries via a one-step protocol and utilized JAMP for subsequent sequence analysis.

Their findings indicate that the limited read length of Illumina sequencing hinders haplotype resolution, as shorter sequences eliminate diagnostic nucleotides. They also discovered that 24-64 PCR cycles recovered 14 assigned ZOTUs, with only 12 ZOTUs having a 100% match to Sanger sequences. The authors conclude by suggesting that: 1) longer read approaches might enable better haplotype detection; 2) the primers used in this study effectively detected targeted species, but alternative primer sets should be tested for potential variations in results; 3) the impact of increased PCR cycles on the relative abundance of quality-passed reads in low-diversity mock samples needs further examination to determine whether this observation extends to more diverse communities; and 4) contamination between libraries and tag-jumping can lead to erroneous read-to-sample assignments, necessitating careful consideration in the development of laboratory protocols and bioinformatics pipelines.

While appreciating the effort put into this emerging area of research, the study's purpose is unclear, as DNA metabarcoding for haplotype population genetic analysis is not particularly novel. Additionally, the influence of DNA concentration and PCR cycles on haplotype recovery has been previously explored. The paper's focus on future work and improvements raises concerns about its immediate relevance.

In a real community sample, bioinformatic filtering strategies and haplotype mock communities can assist in data cleaning decisions, making the analysis of DNA concentration and PCR cycles less relevant. Furthermore, qPCR for tissue samples could provide more information about naturally occurring DNA concentrations and primer bias. The study's generalizability is also questionable without testing a real environmental sample.

To enhance the study's usefulness for a broader audience, it could be reframed to offer a more cost-effective method for studying bulk samples of species compared to Sanger sequencing. This approach could move away from whole community assessments and provide valuable insights for researchers in the field.

Reviewer #3: The manuscript by Serrana and Watanabe presents a study to explore metabarcoding as a tool for extracting haplotype sequences for genetic diversity assessment in freshwater systems. The results show successful application of DNA metabarcoding for this purpose on a small scale mock community study, and also highlights the challenges in bringing this approach to environmental scale. The subject of this manuscript is highly relevant, and the results were generated and analysed thoughtfully and thoroughly. The authors describe the potential and some of the challenges associated with the DNA-based haplotype analysis approach, but little discussion is paid to the path forward. My greatest concern with the study is the size and diversity of the utilised mock community, and its ability to highlight the possibilities of using metabarcoding for this purpose, but I recognise that the results look promising, and can be used carefully.

Specific comments:

The mock community is small (4 species, total specimens = 20), and has low taxonomic diversity (3 stonefly and 1 mayfly species). Why did the authors decide on these species? Increased taxonomic diversity in the mock community as well as number of species would have added to the robustness of the results generated.

L129: Which sequencing kit was used?

L162: Truncating reads to 421 bp merged some of the haplotypes. Was this an expected result? Did the authors perform an in-silico primer analysis based on the known haplotypes? Could the use of a different DNA barcode have prevented this?

L188: Some of the expected haplotypes were not detected at all. The authors do not discuss this result further. It would be nice to have some discussion about how to proceed with overcoming this challenge.

L218: The presented statistics about Nanopore data are out of date, expectations of data quality have improved since chemistry R9.

L228: The authors state that eliminating false or artificial haplotypes will be difficult, but is there any idea about how this challenge could be addressed going forward?

The discussion could be improved further by the inclusion of a closing paragraph that sums up the perspectives and recommendations to develop the approach further toward implementation on environmental samples used freshwater assessments.

6. PLOS authors have the option to publish the peer review history of their article (what does this mean?). If published, this will include your full peer review and any attached files.

Reviewer #1: No

Reviewer #2: No

Reviewer #3: No

---

## [Author Response · Author response to Decision Letter 0]

24 May 2023

*Please refer to the "Response to Reviewers" section below.

---

## [Decision Letter · Decision Letter 1]

11 Jul 2023

Haplotype-level metabarcoding of freshwater macroinvertebrate species: a prospective tool for population genetic analysis

PONE-D-23-02290R1

Dear Dr. Watanabe,

I apologize for the delay.

We’re pleased to inform you that your manuscript has been judged scientifically suitable for publication and will be formally accepted for publication once it meets all outstanding technical requirements.

Kind regards,

Claudia Isabella Pogoreutz

Academic Editor

PLOS ONE

Additional Editor Comments (optional):

Reviewers' comments:

Reviewer's Responses to Questions

**Comments to the Author**

1. If the authors have adequately addressed your comments raised in a previous round of review and you feel that this manuscript is now acceptable for publication, you may indicate that here to bypass the “Comments to the Author” section, enter your conflict of interest statement in the “Confidential to Editor” section, and submit your "Accept" recommendation.

Reviewer #1: All comments have been addressed

2. Is the manuscript technically sound, and do the data support the conclusions?

Reviewer #1: Yes

3. Has the statistical analysis been performed appropriately and rigorously? 

Reviewer #1: Yes

4. Have the authors made all data underlying the findings in their manuscript fully available?

Reviewer #1: Yes

5. Is the manuscript presented in an intelligible fashion and written in standard English?

Reviewer #1: Yes

6. Review Comments to the Author

Reviewer #1: (No Response)

7. PLOS authors have the option to publish the peer review history of their article (what does this mean?). If published, this will include your full peer review and any attached files.

Reviewer #1: No

---

## [Editor Report · Acceptance letter]

14 Jul 2023

PONE-D-23-02290R1 

Haplotype-level metabarcoding of freshwater macroinvertebrate species: a prospective tool for population genetic analysis 

Dear Dr. Watanabe:

I'm pleased to inform you that your manuscript has been deemed suitable for publication in PLOS ONE. Congratulations! Your manuscript is now with our production department. 

Kind regards, 

on behalf of

Prof. Claudia Isabella Pogoreutz 

Academic Editor

PLOS ONE